# Divergent Resistance Mechanisms to Immunotherapy Explain Responses in Different Skin Cancers

**DOI:** 10.3390/cancers12102946

**Published:** 2020-10-13

**Authors:** Emmanuel Dollinger, Daniel Bergman, Peijie Zhou, Scott X. Atwood, Qing Nie

**Affiliations:** 1Department of Mathematics, University of California, Irvine, CA 92697, USA; edolling@uci.edu (E.D.); drbergma@uci.edu (D.B.); peijiez1@uci.edu (P.Z.); 2Department of Developmental and Cell Biology, University of California, Irvine, CA 92697, USA; 3NSF-Simons Center for Multiscale Cell Fate Research, University of California, Irvine, CA 92697, USA; 4Center for Complex Biological Systems, University of California, Irvine, CA 92697, USA; 5Chao Family Comprehensive Cancer Center, University of California, Irvine, CA 92697, USA

**Keywords:** immunotherapy, single-cell transcriptomics, biomarkers, cell–cell communication, mathematical oncology

## Abstract

**Simple Summary:**

Despite the successes of immune checkpoint therapy in treating metastatic skin cancers, most patients either fail to respond or become unresponsive to immunotherapy. We found that the cell–cell communication of two different immune cell types, macrophages and memory B cells, correlate very strongly with the response to immunotherapy. We built a mathematical model based on these results that predict different skin cancers would have different response rates and predict that a high ratio of memory B cells to macrophages is optimal for a response to immunotherapy.

**Abstract:**

The advent of immune checkpoint therapy for metastatic skin cancer has greatly improved patient survival. However, most skin cancer patients are refractory to checkpoint therapy, and furthermore, the intra-immune cell signaling driving response to checkpoint therapy remains uncharacterized. When comparing the immune transcriptome in the tumor microenvironment of melanoma and basal cell carcinoma (BCC), we found that the presence of memory B cells and macrophages negatively correlate in both cancers when stratifying patients by their response, with memory B cells more present in responders. Moreover, inhibitory immune signaling mostly decreases in melanoma responders and increases in BCC responders. We further explored the relationships between macrophages, B cells and response to checkpoint therapy by developing a stochastic differential equation model which qualitatively agrees with the data analysis. Our model predicts BCC to be more refractory to checkpoint therapy than melanoma and predicts the best qualitative ratio of memory B cells and macrophages for successful treatment.

## 1. Introduction

Checkpoint immunotherapy can drive durable responses in many metastatic cancers, with most adverse events being grades 1 or 2 [1,2,3,4]. Current FDA-approved checkpoint inhibitors fall into two categories: cytotoxic T-lymphocyte-associated protein 4 (CTLA-4) inhibitors and programmed cell death protein 1 (PD-1) and PD-1 ligand 1 (PD-L1) inhibitors. CTLA-4 expressed by T regulatory cells (Tregs) outcompete costimulatory molecules on cytotoxic T lymphocytes (CTLs) necessary for their activation, which results in anergy and eventual apoptosis. Cancer and immune cells express PD-L1, which binds to PD-1 expressed by effector cells including CTLs and also by innate immune cells such as natural killer (NK) cells [5]. Binding of PD-1 on effector cells inhibits their cytotoxicity and also promotes anergy and eventual apoptosis [1]. The inhibition of either pathway leads to durable cancer regression in many cancers with varied somatic mutations [1,3]. Checkpoint therapy’s utility remains limited, however, with most patients either not responding or acquiring resistance to treatment [3].

Despite the promise of cancer checkpoint immunotherapy, our understanding of how these therapies affect a system as responsive and dynamic as the immune system remains incomplete. Many studies focus on the effect of checkpoint therapy on CTLs [1,6,7,8,9,10], and the direct downstream effects of checkpoint therapy on other cells that express PD-1, such as macrophages and B cells, remains understudied. Notably, two major recent studies sequenced the transcriptome of the tumor microenvironment (TME) at a single-cell level before and after checkpoint therapy in melanoma [9] and in basal cell carcinoma (BCC) [8], and both these studies focused on the effect of checkpoint therapy on CTLs. However, the effects of checkpoint therapy on different immune cell types have been previously observed [10,11,12]. In a phase I clinical trial for nivolumab, divergent and even opposite effects of nivolumab on T cells and B cells were observed [11]. More recently, B cells have been shown to correlate with response to checkpoint immunotherapy even more strongly than CTL presence [10,12]; however, this remains contentious, with other studies showing no effect [13]. Macrophages have also been identified as critical components of resistance to immunotherapy [14,15,16]. More broadly, there is a lack of understanding of how these different cell types communicate with one another in the TME, particularly in the context of immunotherapy. Cellular communication and coordination have been recently highlighted in processes such as wound healing [17] and communication between different immune cell types have been well-characterized in different contexts.

Since the potential of single-cell RNA-sequencing (scRNA-seq) was demonstrated for the first time on blastomeres in 2009 [18], the ability to partially capture the transcriptome of individual cells has driven insight in many disparate areas of research, including understanding myoblast differentiation [19], identifying rare cancer populations [20], and others [21,22]. scRNA-seq is particularly well-suited to holistically analyze different immune cell types, due to its ability to capture high-resolution transcriptomic data from many cell types at once. Recently, there has been a surge in methods that infer cell–cell communication from transcriptomic data, allowing the discovery of communication between cell types that were previously experimentally challenging. Dynamical systems modeling has successfully modeled the TME [23,24,25] (reviewed in [26]) and can be parameterized by scRNA-seq data analyses to explore the roles of regulations and predict responses to immunotherapy, pointing ways to new therapeutic interventions.

To compare and contrast the immune responses of responders and non-responders to checkpoint therapy, we chose to analyze two scRNA-seq datasets from BCC and melanoma patients treated with checkpoint inhibitors [8,9]. These cancers have inherent differences: melanoma is a highly immunogenic disease that is prone to metastasize quickly, whereas BCC is lowly immunogenic and rarely metastasizes [8,9]. These stark differences allowed for a robust comparison of the differences in response to immunotherapy. We found that memory B cells are most present in responders and vis-versa for macrophages. We characterized their cellular signaling and found that macrophages strongly inhibit memory B cells in melanoma non-responders. However, the immune inhibitory signaling increases in responders in BCC non-responders, along with a strong increase in PD-1 signaling. To fully explore the dynamics of the system, we built a three-state dynamical continuum model that predicts the responsiveness to immunotherapy rests on a high ratio of B cells to macrophages pre-treatment. In addition, the model predicts that BCC will be less responsive to therapy than melanoma, and that an insufficient dose of immunotherapy could induce cancer progression.

## 2. Results

### 2.1. BCC and Melanoma Exhibit Similar Responses to Checkpoint Immunotherapy

To characterize differences between responders and non-responders after checkpoint immunotherapy, we analyzed two scRNA-seq datasets from melanoma [9] and BCC [8] patients before and after immunotherapy. The melanoma scRNA-seq dataset consists of 48 FACS-sorted CD45+ samples (i.e., all the immune cells were sorted) from 32 patients with metastatic melanoma before and after either anti-PD-1, anti-CTLA-4, or combination treatment. The BCC dataset consists of 24 site-matched samples from 11 patients with metastatic or locally advanced BCC before and after PD-1 blockade. We clustered the immune cells from each dataset separately and found they both contain CD4+ T cells, CD8+ T cells, T regulatory cells (Tregs), macrophages, memory B cells, plasma B cells, plasmacytoid dendritic cells (pDCs), and cycling T cells (Figure 1A,B; Appendix A). Overall, our clustering recapitulated the original analysis [8,9] (Appendix A). We then calculated the fraction of cells from responders and non-responders in each cell type by cancer and found that the overall percentage of responders and non-responders in each cluster was similar across both cancers (Figure 1C,D). Between BCC and melanoma, CD8+ T cells consistently showed roughly equal distribution between responders and non-responders, whereas memory B cells were highly concentrated in responders and macrophages were highly concentrated in non-responders. To determine whether these trends are generalized or patient-specific, we compared the percentage of macrophages, memory B cells, CD8+ T cells and Tregs in all immune cells per patient and compared responders to non-responders (Figure 1E–H). We found that macrophages represent a higher percentage of cells per patient in non-responders and that the opposite is true for memory B cells. The percent of CD8+ T cells and Tregs were similar in responders and non-responders. Overall, the distribution of immune cells in responders and non-responders show remarkable similarity in both cancers, despite the differences in immunogenicity and sequencing technologies used for each cancer.

There is a growing body of literature indicating that mutations in common oncogenes such as *BRAF* correlates with poor prognosis for melanoma [27,28]. To determine if *BRAF* mutations had an effect on the proportion of memory B cells or macrophages, we compared the percentage of these cell types from patients with and without *BRAF* mutations [9] and the percentage of these cell types in responders and non-responders with *BRAF* mutations (Appendix A). Although not significant due to the small cohort size, the trends are consistent with our comparison in the full cohort where a smaller ratio of B cells to macrophages are seen in non-responders and *BRAF* mutant patients (Appendix A).

### 2.2. Memory B Cells Are More Active in Post-Treatment Responders and Anergic in Post-Treatment Non-Responders

As memory B cells were highly concentrated in immunotherapy responders in both datasets and may provide insight into mechanisms by which patients respond, we subclustered the memory B cells in both datasets and found the melanoma memory B cells to be well-mixed with regards to treatment, response, and patient, whereas the BCC memory B cells suffer from batch effects stemming from the small patient size (Figure 2A,B; Appendix A). When comparing the memory B cell subclusters between BCC and melanoma, we observe differences in gene expression unique to each cancer (Appendix A) suggesting the memory B cells are not occupying similar states and may be differentially interacting with the TME.

With the increase in Memory B cell complexity, we used similarity matrix-based optimization method (SoptSC) (Nie, Irvine, CA, USA) to infer their lineage [29]. The melanoma and BCC memory B cell lineages show distinct trajectories that reflect the differences in cellular states between the two cancers (Figure 2C,D). However, when segregating the pseudotime trajectory along activation scores that reflect memory B cells binding to their specific antigen and actively expressing costimulatory receptors for T helper 1 (Th1) cells [30], both lineages show an increase in activation score at their terminus (Figure 2C,D). Both responders and non-responders show the increase in activation at the trajectory terminus, suggesting that the immune system is attempting to activate memory B cells in distinct ways within each cancer.

To further define how memory B cells are interacting with their environment, we developed a score for memory B cell anergy to go along with the activation score (Appendix A, Methods). If activated B cells don’t receive costimulatory signals from Th1 cells, they become anergic, non-responsive to stimulation, and eventually apoptose [30]. The average normalized expression of each set of genes that make up activation or anergy scores were calculated for each cell and stratified on pre- or post-treatment and response of the patient. In the melanoma dataset, the activation score for pre-treatment responders is significantly lower than in post-treatment responders, and the activation score is significantly higher in pre-treatment non-responders than in pre-treatment responders (Figure 2E), which makes sense given memory B cells should be more active in responders after treatment and not in non-responders. The only significant difference in the anergy scores for melanoma patients comes from the post-treatment non-responders, which are more anergic than those in pre-treatment (Figure 2G). BCC memory B cells show similar, but not significant, trends in activation and anergy to memory B cells in melanoma (Figure 2F–H).

### 2.3. Macrophages in BCC Have a Pro-inflammatory Genotype, Regardless of Responder Status

Macrophages have important roles in cancer immune suppression and correlate with poor prognosis, with drugs being developed to inhibit their suppressive ability [14,15]. In particular, inhibiting the macrophage colony stimulating factor (CSF1) pathway increases the sensitivity of pancreatic ductal adenocarcinoma to checkpoint immunotherapy by decreasing the number of macrophages in the TME, increasing antigen presentation on macrophages, and increasing checkpoint ligands on tumors [16]. To characterize the role of macrophages in resistance to immunotherapy in each cancer, we subclustered the macrophages and found that macrophages are mostly present in non-responders in both cancers (Figure 3A–D; Appendix A). Similar to memory B cells, when comparing the macrophage subclusters between BCC and melanoma, we observe differences in gene expression unique to each cancer (Appendix A) suggesting the macrophages are not occupying similar states and may be differentially interacting with the TME.

We built macrophage inflammatory scores using genes that are either defined as “pro-inflammatory” or “anti-inflammatory” in Gene Ontology (Appendix A) [31,32]. We chose not to emphasize the classical M1/M2 paradigm due to the increasing amount of evidence indicating that macrophages show more heterogeneity than two states, although we should note that some of these markers are included in our scoring scheme, and the M1 and M2 scores showed comparable trends (Appendix A) [33,34,35,36]. In the melanoma dataset, we found that anti-inflammatory gene expression correlates well with the percentage of macrophages found in post-treatment non-responders, indicating that macrophages in the melanoma TME are involved in the refractory response to immunotherapy (Figure 3E). However, BCC macrophages have very low expression of anti-inflammatory genes in all backgrounds, suggesting that BCCs may not regulate immunotherapy response by inflammatory signals (Figure 3F). Although both cancers have more macrophages in non-responders, the macrophages have unique inflammatory signatures that are linked to different processes (Appendix A). Using SoptSC to generate a lineage for macrophages, we observed distinct trajectories that reflect the differences in cellular states between the two cancers but a similar increase in anti-inflammatory scores at the trajectory terminus (Figure 3G,H).

To understand how our results relate to the immune system in the TME of metastatic and primary melanoma tumors, we re-analyzed a third dataset of either untreated or non-responsive patients with well-defined tumor sites [37]. We found that the macrophages in the primary tumors express more anti-inflammatory genes than macrophages in the metastatic site; however, these results were mostly not significant due to the small amount of data available (Appendix A).

### 2.4. Anti-Inflammatory Signaling Is Reduced in Melanoma Responders and Increased in BCC Responders

To correlate changes of intra-immune signaling between B cells and macrophages with immunotherapy response, we used SoptSC to construct probabilistic cell–cell signaling interactions. Signaling probabilities are quantified based on the weighted expression of signaling pathway components between sender-receiver cell pairs inferred through the expression of ligand–receptor pairs and their downstream targets (Methods) [29]. We subsetted memory B cells, plasma B cells, and macrophages in both datasets and calculated the probability of cluster-cluster cell signaling, which averages individual cell signaling probabilities within each cluster (Figure 4A,E). We included the plasma B cells in the analysis because of the stark difference in the fraction of responders and non-responders between the two cancers (Figure 1C). We chose three pathways of study: Fc fragment of IgG receptor IIb (FCGR2B), interleukin 6 (IL6), and PD-1. FCGR2B is a well-characterized inhibitory pathway used by macrophages to inhibit B cells [38]. The IL6 pathway has been correlated with B regulatory cell (Breg) activation, which has been implicated in many immunological tolerance mechanisms such as organ transplantation [39], cancer [40], and self-stimulation of tumor cells [41]. The PD-1 pathway is used as a control for the response. We chose these pathways because of their biological relevance, their downstream targets are well-characterized, and the genes in each pathway were present in both datasets.

In the melanoma dataset, we found that the FCGR2B pathway is strongly upregulated in non-responders (Figure 4B). The majority of the FCGR2B-mediated inhibition goes from macrophages to memory and plasma B cells, suggesting that B cells are selectively inhibited in non-responders. Concurrently, the IL6 pathway is upregulated in non-responder memory B cells, with signaling directed towards the macrophages and plasma B cells (Figure 4C), suggesting an anti-inflammatory response in these cells and further suppression in the immune response. PD-1 signaling is increased in responders, with the source of signaling switching from macrophages in non-responders to plasma B cells in responders (Figure 4D). These results match reports that anti-PD-1 therapy response tends to correlate with PD-1 expression [11].

In the BCC dataset (Figure 4E), we subsetted macrophages, both B cell types, and cancer-associated fibroblasts (CAFs), which have high IL6 signaling. FCGR2B occurs with similar strength in responders and non-responders, with a switch from memory B cell inhibition in non-responders to plasma B cell inhibition in responders (Figure 4F). IL6 signaling increases in responders, with the majority of the signaling coming from CAFs and going to plasma B cells in responders (Figure 4G). In non-responders, the signaling still comes from CAFs but now signals mainly to macrophages (Figure 4G). The upregulation of IL6 with checkpoint blockade was previously characterized in melanoma mice models but not to this current resolution [42]. Finally, the PD-1 signaling is drastically upregulated in responders, with the majority of the signaling directed towards memory B cells in BCC instead of plasma B cells in melanoma (Figure 4H). These results, especially the trends of the FCGR2B and IL6 pathways, indicate that the immune system in melanoma is actively inducing an immune-suppressive environment, which is contributing to resistance; however, BCC seems to only induce a suppressive environment in responders, indicating that there is a different mechanism of resistance, relying on the simple lack of sufficient activation of immune cells during therapy.

### 2.5. A Dynamical Model on Interactions Among Memory B Cells, Macrophages and Skin Tumors

To better understand the dynamics of the immune system during treatment and specifically predict the best immune cell composition for a response, we developed a three-state continuum dynamical model based on the bioinformatic clustering, lineage, and cellular signaling analyses. We chose cancer, B cells, and anti-inflammatory macrophages that inhibit the activation of immune cells (referred to as simply “macrophages”) as our state variables (Figure 5A and Methods). The cancer undergoes logistic growth and has four possible steady states: none, low (~10^3^ cells), high (~10^8^ cells), and very high (~10^9^ cells). The B cells kill cancer cells and macrophages inhibit B cell proliferation. The parameters of the dynamical model were selected based on our bioinformatic analyses and previous literature (Appendix A), were non-dimensionalized to simplify our analysis (Appendix A) and the equations for the dynamical model are shown in the methods.

We made two assumptions from the signaling analyses that differentiate the divergent refractory mechanisms of each cancer. First, consistent with macrophages in BCC having less of an anti-inflammatory phenotype (Figure 3G,H), we assumed that the cancer-mediated up-regulation of macrophage proliferation is weaker in BCC relative to melanoma. Second, consistent with stronger B cell suppression in BCC (Figure 4B,C,F,G), we increased the negative regulation of B cells by BCC cancer cells relative to that in melanoma (all parameters are shown in Appendix A).

To understand the possible dynamics predicted by the model without immunotherapy, we computed several representative trajectories (Figure 5B). We fixed the starting immune populations and each parameter (Appendix A) and varied the initial cancer burden between low, medium, and high. In both melanoma and BCC, the high cancer burden remained high while the immune populations followed different trajectories. Macrophages steadily increased in melanoma and remained low in BCC, in accordance with previous observations [43,44]. Both melanoma and BCC were not able to transition from a low cancer burden to a high one in the chosen set of parameters. The medium cancer burden regressed to a low cancer burden only in melanoma while accompanied by a transient spike in memory B cells, whereas the medium cancer burden in BCC progressed to a high cancer burden.

### 2.6. The Model Predicts the Most Likely Immune Cell Composition for Responders and Shows BCC Is Less Likely to Respond to Treatment

To understand the effects of immunotherapy on cancer burden, we analyzed the steady states of our model within certain biologically relevant parameter ranges (Appendix A). Overall, our system displayed multi-stability, a common concept in cancer state modeling [23], in both melanoma and BCC: the system could evolve towards two or more steady states depending on the level of cancer burden (Figure 5C).

We decided to vary the killing rate of B cells k as a proxy for immunotherapy, and study how the steady states change as we increased k (i.e., bifurcation analysis). We found that melanoma and BCC responded similarly to immunotherapy (Figure 5D). We observed responders in both cancer backgrounds where a very high cancer burden transitioned to a low cancer burden. In melanoma responders, an increase in B cells and a large decrease of macrophages was observed. We saw the same pattern in the immune profile of BCC, except the increase in B cells was larger while the decrease in macrophages was smaller. On the other hand, non-responders showed a small decrease in B cells and an increase in macrophage population, potentially up to several orders of magnitude in the melanoma case.

To predict responsiveness to immunotherapy, we determined the immune cell composition for responders and non-responders, pre-treatment. We compared the equilibrium number of macrophages and B cells just before and just after the transition from non-responders to responders as we increased the B cell killing rate k (Figure 5D). Our model predicted that responders pre-treatment have a high B cell/macrophage ratio: in order to be a responder, the initial killing rate of B cells needs to be relatively higher, which in our bifurcation analysis implies a higher amount of B cells and a lower number of macrophages. Indeed, this prediction matched well with ratios calculated from the data (Appendix A). In melanoma responders pre-treatment, there was a much higher ratio of B cells to macrophages compared to the ratio in non-responders pre-treatment. The ratio of melanoma non-responders decreased post-treatment, which also matched the model results. These predictions matched the B cell/macrophage ratios in BCC during treatment, suggesting that predicting BCC response to immunotherapy may be possible during the early stages of treatment and that other biomarkers may be necessary to predict BCC response pre-treatment.

In our chosen parameter regime, the value of the immunotherapy killing rate at which a patient would become a responder was lower for melanoma than BCC (Figure 5D). This relationship between the two cancers persisted even as we varied d_e_ (i.e., the death rate of B cells) leading us to predict melanoma to be more likely to respond to immunotherapy than BCC (Appendix A).

### 2.7. Noise-Induced Cancer Progression and Regression Potentially Account for Therapy-Resistance in BCC

In the highly complex cancer-immune interacting environment, fluctuations in cell populations may induce random transitions among meta-stable states [25,45]. We therefore incorporated stochastic effects into our three-component dynamical model (equations detailed in the methods) (Appendix A). In our stochastic model, the inclusion of random fluctuations in cell population dynamics allows for spontaneous (as opposed to by varying a parameter) transitions between cancer states with various burdens, contributing another source to affect the checkpoint therapy outcome by the spontaneous progression or regression of cancer, which has been noted in previous studies [46,47,48].

In order to compare the relative stability of noisy cancer states, we constructed a cancer-state landscape to visualize the global structures of attractor basins in melanoma and BCC populations and their transition dynamics (Figure 6A). The less likely cancer states correspond to shallower basins in the landscape—the intuition here is analogous to the classic Waddington landscape for cell fate commitment [49], or wells in activation energy barrier diagrams. The deeper the well, the higher the energy required and the less likely the transition to a different well becomes. The cancer-state energy landscape agrees with our bifurcation analysis by showing two connected energy wells representing “stable” cancer states with “low” and “high” tumor burdens. The connectivity between these cancer wells suggests that spontaneous transitions can occur in both cancer types, corresponding to tumor progression and regression. We also observed that the cancer well of the low-burden state in BCC is shallower than in melanoma and the high-cancer state in BCC is deeper than in melanoma, suggesting a higher probability to transition to the higher-burden state and a smaller probability of the reverse transition (matching our prediction of BCC response from Figure 5) (Figure 6A).

A unique feature of stochastic vs deterministic (e.g., the model represented in Figure 5) systems is the possibility of a transition between stable states. The specific transition path the system follows can discriminate between a growing and regressing cancer. To study these transition paths, we implemented the geometric minimal action method (gMAM) which determines the likelihood of each path (Methods) [50]. When melanoma transitioned from a high cancer state to a low cancer state (i.e., regresses) there was a strong increase in B cells, which was not true of the reverse transition (Figure 6B). In BCC, there is a similar pattern in the B cell population, though it is less pronounced.

To quantify how checkpoint therapy affects the likelihood of spontaneous tumor progression and regression, we calculated the change in activation energies between the two cancer states as the killing rate is increased (Figure 6C). Comparing these two curves, melanoma exhibited greater sensitivity to therapy with the activation energy decreasing more quickly. However, both cancers exhibit a surprising characteristic: the activation energy for regression initially decreases in the bistable region before growing at the higher end of this region. This indicated that a failure to push the system into a state with a unique attractor—a single, low cancer burden one—could make cancer less likely to spontaneously regress. We found that the barrier height for regression in BCC is generally larger than in melanoma with similar killing rates, predicting BCC patients to be more refractory to immunotherapy in general.

When we quantified the activation energy for progression, we first observed that it was higher for melanoma than BCC, indicating a higher propensity for melanomas to have a durable response. We also noted that in BCC, the activation energy for progression is more sensitive to immunotherapy. At lower values of the killing rate k, therapy drove this barrier down making it more likely for an initial response to be reversed. This may provide a potential explanation for the unsatisfactory outcome of checkpoint therapy in BCC [47,48,51,52,53]. However, this trend eventually reverses and at higher killing rates, therapy makes spontaneous progression less likely.

## 3. Discussion

Melanoma is a relatively rare and very dangerous immunogenic disease that arises from neural crest cells, whereas BCC is a very common and relatively benign non-immunogenic disease that arises from stem cells of the skin and hair follicle. However, our data suggest that their immune cell composition between responders and non-responders to immunotherapy is similar, albeit for different reasons. Melanoma-associated macrophages in non-responders seem to be more anti-inflammatory, suggesting that macrophages may be an important resistance mechanism to immunotherapy as suggested in pancreatic cancer [16]. BCC-associated macrophages seem to be more pro-inflammatory, suggesting they are not important to immunotherapy resistance and that the barrier to BCC response to immunotherapy is a matter of immune cell recruitment and activation, not overcoming resistance. This matches well with reports that there is a sharp increase in immune cells after checkpoint therapy [8].

Our bifurcation analysis indicated that, dependent on the individual sensitivity toward the therapy in increasing the cancer-killing rate k of B cells, the patient may either have a durable response, a partial response or a refractory response. These results could explain why some patients appear to have a naturally acquired resistance to immunotherapy [54]. Our model suggests a high memory B cell count and low macrophage count (relative to each cancer) would indicate a likely response.

Despite the similarities in cell composition in each cancer, we also found important differences in the dynamics of melanoma and BCC cancers. From our energy landscapes, we observed a shallow low cancer burden well in BCCs, suggesting BCCs have a higher probability to transition to a higher cancer burden than melanoma. Our analysis of activation energies additionally suggests that BCC is less likely to respond to checkpoint therapy and the likelihood of post-therapy cancer recurrence is higher than in melanoma. BCC’s resistance to immunotherapy seems to be borne out in the literature, although this is still under investigation [47,48,51,52,53]. In fact, the model suggests that an insufficient dose of immunotherapy could have adverse effects for some BCC patients with a low pre-therapy killing rate, increasing their risk of tumor progression.

A crucial assumption we have made throughout this study is that memory B cells are directly affecting the cancer, either by releasing pro-inflammatory cytokines or by antibody production. Unfortunately, we were unable to verify whether these memory B cells were producing more antibodies in responders from the scRNA-seq datasets. Furthermore, it is unclear why memory B cells are more implicated in this response than plasma B cells. Memory B cells are known to produce antibodies with higher affinities compared to plasma B cells [55], but require periods where they are not stimulated to properly mature, perhaps implying that the level of activation of B cells in responders before treatment needs to be relatively lower, at least for a period of time. Indeed, this intuition matches well with our results that memory B cells in responders pre-treatment are less activated than in non-responders pre-treatment.

## 4. Materials and Methods

### 4.1. Clustering

All analyses unless otherwise noted was the same for all datasets. The UMI count matrix for [9] was provided by personal communication from the authors. The UMI matrix from [8,37] were downloaded via GEO, accession GSE123813 and GSE115978 respectively. No additional human clinical trials were performed for the preparation of this manuscript; written informed consent was given for all studies and can be found in the method sections of each paper.

We excluded all cells with counts of less than 200. The UMI counts were normalized and scaled using the SCTransform tool in Seurat v3 (Satija, New York City, NY, USA) [56,57]; briefly, gene expression was normalized by taking the residuals of a generalized linear model that fits the counts of each gene across cells to a “regularized” negative binomial regression, with covariate cell sequencing depth. In this GLM, the Pearson residuals are the scaled gene expression values and were used for downstream analyses. These scaled gene values were used as input to PCA. The resulting first 30 dimensions of the PCA were used to generate the UMAP projections, with default parameters. The 30 first dimensions of the PCA were also used to calculate the shared-nearest neighbor network, which was used to cluster the cells (the smart local moving algorithm and resolution = 0.3 was used for clustering for both datasets; all other parameters were left as default).

To identify clusters, the differential expression on each cluster was performed (Wilcoxon Rank Sum test; aside from thresholding the minimum fraction of cells that need to express a gene for that gene to be included to 0.25, all parameters were set to Seurat default) and the resulting top 50 differentially expressed genes were supplied to Enrichr (Ma’ayan, New York City, NY, USA), a gene list enrichment analysis tool [58]. Clusters were identified by holistically considering different datasets (e.g., Human Gene Atlas, Mouse Gene Atlas, ARCHS4 Tissues and ARCHS4 Cell-lines).

To facilitate comparison across datasets, the T cell clusters were grouped by expression of CD8+ and/or CD4+; Tregs were identified by Enrichr and FOXP3+ expression. The B cells in the melanoma dataset were identified by Enrichr (plasma B cells) and specific markers (MS4A1 and CD40 for memory B cells). The B cells in the BCC dataset were identified by expression of the top differentially expressed genes between the memory B cells and plasma B cells in the melanoma dataset (plasma B cells: MZB1, IGHGP, IGHG3, IGHG1; memory B cells: CD79A, CD19, BANK1, IGHM, MS4A1).

The dataset from [37] was analyzed using the same pipeline. The original cell labels from the paper were used for cluster identification.

### 4.2. Lineage Analysis and Cell–Cell Signaling Inference

The lineage analysis and cell–cell signaling was performed in SoptSC (Nie, Irvine, CA, USA) [29]. SoptSC is a similarity matrix-based method for inferring cell lineage and cell signaling. Briefly, SoptSC calculates a cell–cell similarity matrix *S* based on a low-rank representation of the log-transformed UMI count matrix. Our specific procedure for inferring clusters and building the cell lineage graph did not deviate from that laid out in [29]: the similarity matrix was computed and the clusters and the number of clusters were inferred. For the non-responder subset of macrophages, memory B cells and plasma cells from BCC patients (Figure 4C–E), the memory B cell cluster was manually defined based on their identities in the full dataset.

SoptSC calculates the probability that cells are signaling given a user-defined pathway of {Ligand, Receptor, Downstream upregulated target}. The cluster–cluster signaling graphs were generated by calculating the weighted graph of the cell–cell graph from the probability of signaling. Three pathways were considered: {FCGR2B, CD79A, FAS} and {FCGR2B, CD79B, FAS} were considered (i.e., calculated separately, then averaged) for macrophage-specific inhibitory signals, {PDL1, PD1, BATF} and {PDL2, PD1, BATF} were considered for PD1 signaling, and {IL6, IL6R, FCGR2B} was considered for immune inhibition. The probabilities were calculated in SoptSC by only considering probabilities >0.025 and the resulting probability matrix was visualized in the circlize v0.4.9 package (Brors, Heidelberg, Germany) [30]. The probabilities are relative to the transcriptomic information in each dataset and can only be compared with probabilities in the same dataset.

### 4.3. Heatmaps, Dotplot, Barcharts and Box-and-Whisker Plots

For each dataset, the macrophages were subsetted, imported into SoptSC and clustered as described above. The cluster labels of the subsetted Seurat object were redefined with the SoptSC clusters, and the heatmaps were generated by inputting the specified gene list in the DoHeatmap function of Seurat.

Taking the cluster labels of either the original Seurat clusters (Figure 1) or the cluster labels of SoptSC (Figure 2), the percent of either responder status (Figure 1) or percent of response/treatment (Figure 2) per cluster was calculated by dividing the number of cells in each category by the total number of cells in the cluster.

The percent of cells per patient (Figure 1) was calculated by dividing the number of specified cells (e.g., macrophages) by the total number of cells for that patient (we excluded the patients that had none of the specified cells). The Wilcoxon Rank Sum test was performed using the stat_compare_means function in the ggpubr package v0.2.5, with defaults. The percent of cells per patient for comparison within melanomas with and without *BRAF* and between *BRAF* responders and non-responders was calculated in the same fashion (Appendix A). The mutation status of each patient with whole-exome sequencing is specified in the supplemental materials of [9].

The “activation” and “anergy” scores for memory B cells were calculated by averaging the normalized gene expression for each gene list per responder status. Activation genes: *CD79A*, *IL4R*, *CD40* and *ITGAL*. Anergy genes: *FAS*, *NFKB1*, and *PDCD1*.

The “pro”-inflammation and “anti”-inflammation scores were calculated by averaging the normalized gene expressions for each gene list per cluster. Pro-inflammatory gene list: *CD74*, *CCR7*, *CCL22*, *IFITM1*, *IFITM2*, *IFITM3*, *EREG, CTSB*, *FCN1*, *TNFAIP3*, and *FCER1G*. Anti-inflammatory gene list: *FCGR2B*, *NR1H3*, *A2M*, *CD84*, *VSIG4*, *GPNMB*, *PROS1*, *PLAU*, *APOE*, *PROCR*, *TREM2*, *CD151*, *DAB2*, *NRP1*, *NRP2*, *MMP9*, *SCARB1*, *SPARC*, *ECM1*, *PLXND1*, *ENG*, and *FN1*. Note that the clusters for each cell type were calculated using SoptSC and imported into the Seurat object, see above. The M1 and M2 markers were calculated in the same fashion (from ref [33]) (Appendix A). M1 markers: *CD86*, *ITGAX*, *HLA-DRA* and *STAT1*. M2 markers: *CD163*, *MSR1*, *VEGFA* and *MRC1*.

The gene intersection heatmaps for Appendix A were calculated by doing differential expression on each subset as before and calculating the fraction of genes that were present in each cluster.

### 4.4. Analysis of Immune System in Primary and Metastatic Melanoma

To compare the immune system of primary and metastatic melanoma, we re-analyzed the dataset from [37], GSE115978 (Appendix A). The analysis pipeline was identical to the analysis for [8,9]. The percent of cells per patient was calculated in a similar fashion apart from scaling the fraction of cell types from primary and metastatic sites by the number of patients with primary and metastatic cancer, respectively.

### 4.5. The Three-Component Dynamical Model

Based on the single-cell data analysis, we modeled the dynamics of B cells, macrophages and cancer cells populations, which are emergent from their complex interactions. The assumptions on interactions between B cell and cancer cells are derived from existing literatures (Appendix A). The inclusion of anti-inflammatory macrophages (“macrophages”) and their interactions with other cells constitutes the novel aspect of our work, as most previous work uses pro-inflammatory cells as a third state variable (e.g., [24]). Derived from the single-cell data analysis, the macrophages act to down-regulate B cell proliferation, directly in opposition to the cancer-mediated upregulation of that very process. The macrophages are in turn influenced by cancer and memory B cells by responding to the apoptotic signals from cancer cells as the memory B cells kill them [59].

The model can be expressed in ordinary differential equations (ODEs). We let C, B, and M stand for the state variables of cancer cells, memory B cells, and macrophages, respectively. These three variables are time-dependent. Cancer cells have a proliferation rate a and carrying capacity $b^{-1}$. B cells kill cancer cells at rate k. B cells have a constant influx at rate s and die at rate $d_{e}$. In the presence of cancer, B cells are stimulated and proliferate at a maximal rate $b_{e}$. The cancer mediates this via a Hill function with EC_50_ term $\kappa_{e}$. Macrophages inhibit this proliferation with another Hill function with EC_50_ term $\kappa_{m}$. On the other hand, the cancer can adversely affect the B cell population by encouraging their removal from the system. This happens at a maximal rate of $d_{e}$ and with EC_50_ term $\kappa_{d}$. Finally, macrophages also have a source, g, and death rate, $d_{m}$. Their proliferation can be stimulated by apoptosis of cancer cells as induced by B cell killing, occurring at a maximal rate, p, and with EC_50_ term $\kappa_{a}$. See Appendix A for parameter values and sources.
(1)$$C^{\prime }=aC\left(1-bC\right)-kCB$$
(2)$$B^{\prime }=s-dB+b_{e}\frac{C}{\kappa_{e}+C}\frac{\kappa_{m}}{\kappa_{m}+M}B-d_{e}\frac{C}{\kappa_{d}+C}B$$
(3)$$M^{\prime }=g-d_{m}M+p\frac{kCB}{\kappa_{a}+kCB}M.$$

We remark here that the effects of pro-inflammatory macrophages can be incorporated in this model. The two contributions of these macrophages would be in their plastic conversion with anti-inflammatory macrophages (which are regulated by the tumors) and their upregulation of the B cells. These effects could be equivalently overlapping with the regulation of B cells by tumor cells and can thus be viewed as already accounted for in the model implicitly. Indeed, the proposed model can be viewed as the effective reduction of a four-state model that explicitly includes the pro-inflammatory macrophages, where a formal mathematical analysis is given in Appendix A.

We conducted non-dimensionalization to simplify our analysis (Appendix A). To perform equilibria and stability analysis, we solved the derived fifth-degree polynomial of the steady-state equation and determine the stability using the eigenvalues of the Jacobian (Appendix A). The bifurcation plot can be generated by tracking the change of equilibria with respect to the parameter of interest.

To consider transitions among meta-stable cancer states, we included a time-independent noise term $\sigma \left(X_{t}\right)$ and generated a stochastic differential equation (SDE) model $dX_{t}=b(X_{t})dt+\sigma (X_{t})dW_{t}$, with $b\left(X_{t}\right)$ corresponding to the drift terms in the ODE system, and *W_t_* being a standard Weiner process (Appendix A).

### 4.6. Cancer-State Landscape and Transition Paths

The cancer-state landscape can quantify the relative stability of different meta-stable states perturbed by noise, closely relevant to the notion of the energy landscape, a mathematical realization of Waddington’s epigenetics metaphor [49,60,61]. To generate the landscapes, we simulated a large number of trajectories with randomly chosen initial conditions. Initial conditions were uniformly distributed over the log scale of the state variables. Each subsequent time step was binned based on the 3D coordinates and used to compute the probability a trajectory was in a particular bin. To arrive at the landscapes, we took the marginal probabilities over a given state variable and then computed the negative logarithm to arrive at our potential landscape.

To compute transition paths among meta-stable states, we applied the Freidlin and Wentzell’s (FW) large deviation theory [50], which states that under small noise assumption, the most probable path $\varphi^{*}\left(s\right)$ transiting from state $x_{1}$ to $x_{2}$ corresponds to the minimizer of the action functional.
(4)$$S\left[\varphi \right]=\int_{0}^{T}{\left(\varphi^{\prime }-b\left(\varphi \right)\right)}^{t}D^{-1}\left(\varphi \right)\left(\varphi^{\prime }-b\left(\varphi \right)\right)ds$$
where matrix $D\left(x\right)=\sigma \left(x\right)\sigma^{t}\left(x\right)$ and $\varphi^{*}=\underset{T>0}{inf}\underset{\begin{array}{c} \varphi \left(0\right)=x_{1}\\ \varphi \left(T\right)=x_{2} \end{array}}{inf}S\left[\varphi \right]$.

We set $x_{1}$ and $x_{2}$ as stable fixed points of the ODE system. To tackle the numerical challenges introduced by critical points [62], we implemented a simplified geometric minimal action method (sgMAM) to solve the optimization problem [62]. We used the action functional for these paths to compute the activation energies between stable equilibria. According to the FW theory [50], the larger activation energies indicate longer mean transition time between metastable states.

### 4.7. Code and Data Availability

The data for [9] are stored in dbGAP phs001680.v1.p1, the UMI matrix from [8,37] are stored in GEO, accession GSE123813 and GSE115978 respectively. 

The code used to generate bioinformatics and mathematical results are available upon request.

## 5. Conclusions

Despite immunotherapy significantly advancing cancer therapy and extending patient survival, not much is broadly known about the effects of immune cell communication on patient response. We analyzed and compared two scRNA-seq datasets from melanoma and BCC and found that memory B cells are over-represented in responders, whereas macrophages are over-represented in non-responders. We found that overall inhibitory signaling increased in melanoma non-responders and in BCC responders. These novel results allowed us to build a dynamical continuum model that predicted optimal ratios of memory B cells to macrophages which were validated using the transcriptomic datasets. The model predicted divergent responses to checkpoint therapy for responders and non-responders, as well as differences in immunotherapy response by BCC and melanoma. These predictions point the way towards more personalized and cancer-specific immunotherapy dosing, and establish new immuno-oncology paradigms that enable a better understanding of immunotherapy.

## Figures and Tables

**Figure 1 cancers-12-02946-f001:**
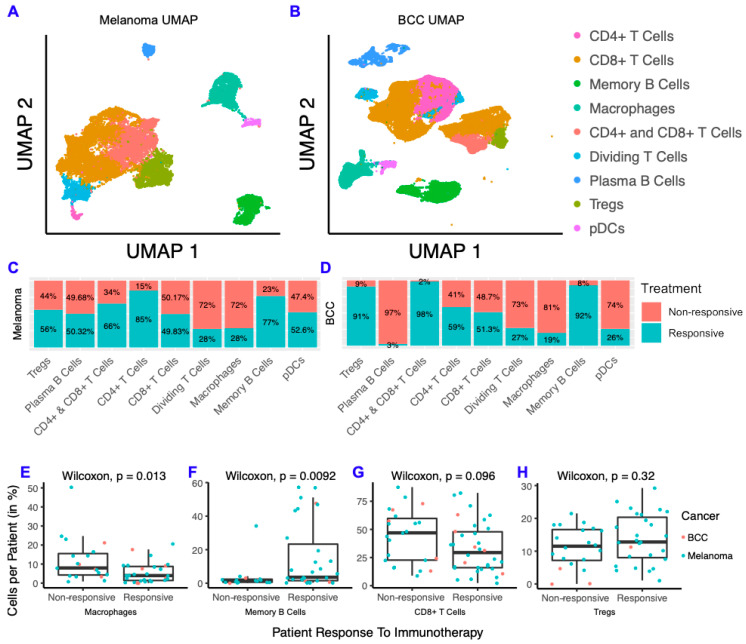
Melanoma and basal cell carcinoma (BCC) have similar responses to immunotherapy. (**A**,**B**) Dimensionality reduction of melanoma (**A**) and BCC (**B**). (**C**,**D**) Distribution of cells from responders and non-responders, grouped by cluster. (**E**–**H**) Percentage of macrophages (**E**), memory B cells (**F**), CD8+ T cells (**G**) and T regulatory cells (Tregs) (**H**) per patient, grouped by responders and non-responders.

**Figure 2 cancers-12-02946-f002:**
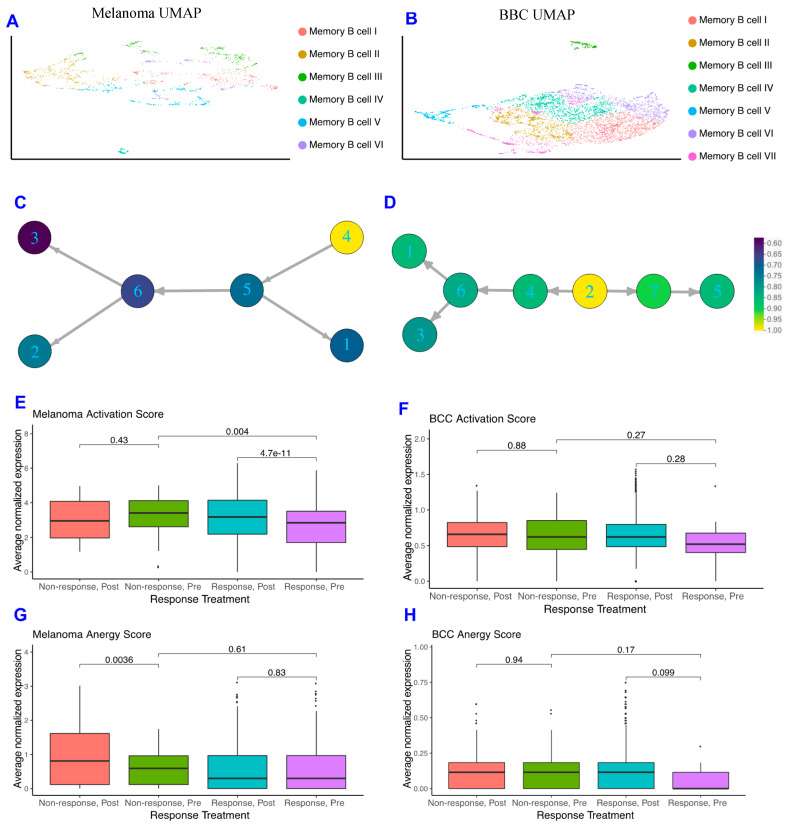
Memory B cells are more activated in non-responders pre response. (**A**,**B**) Dimensionality reduction of the memory B cells subsets of melanoma (**A**) and BCC (**B**). (**C**,**D**) Psuedotime ordering of melanoma (**C**) and BCC (**D**), colored by normalized activation score within each dataset. (**E**,**F**) Activation scores of memory B cells in melanoma (**E**) and BCC (**F**). (**G**,**H**) Anergy scores of memory B cells in melanoma (**G**) and BCC (**H**).

**Figure 3 cancers-12-02946-f003:**
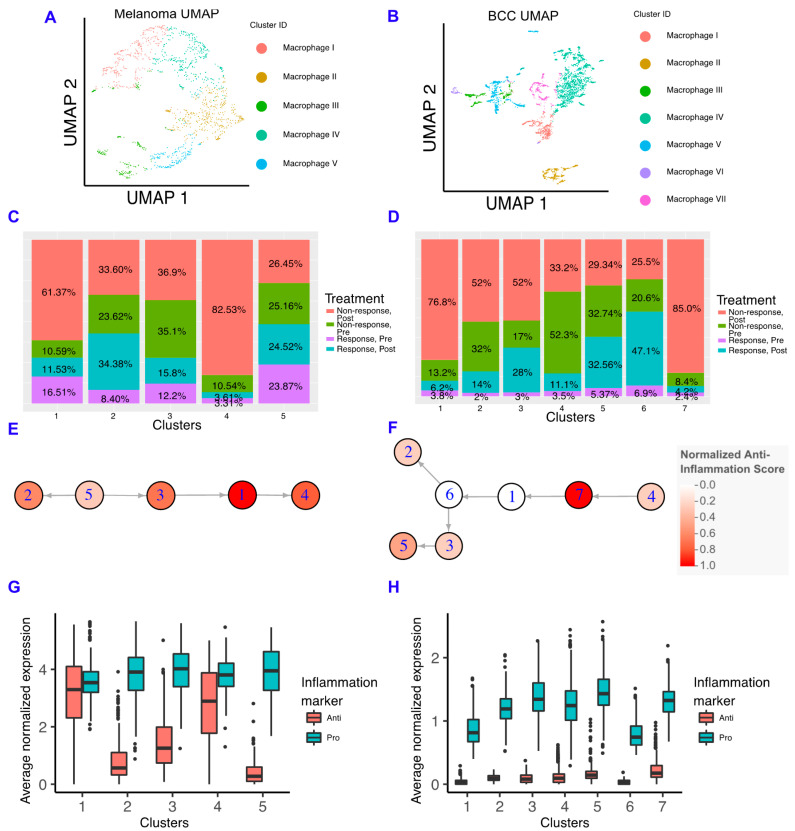
Macrophages in BCC have more of a pro-inflammatory genotype, regardless of responder status. (**A**,**B**) Dimensionality reduction of the macrophage subsets of melanoma (**A**) and BCC (**B**). (**C**,**D**) Percentage of responders/non-responders in pre/post-treatment per macrophage cluster in melanoma (**C**) and BCC (**D**). (**E**,**F**) Average expression of anti- and pro-inflammatory genes by cluster of macrophages in melanoma (**E**) and BCC (**F**). (**G**,**H**) Psuedotime of macrophage clusters in melanoma (**G**) and BCC (**H**). Each psuedotime node is qualitatively colored by a normalized expression of the anti-inflammation score. The melanoma psuedotime correlates well with the percent of non-responders post-treatment, whereas the BCC psuedotime does not.

**Figure 4 cancers-12-02946-f004:**
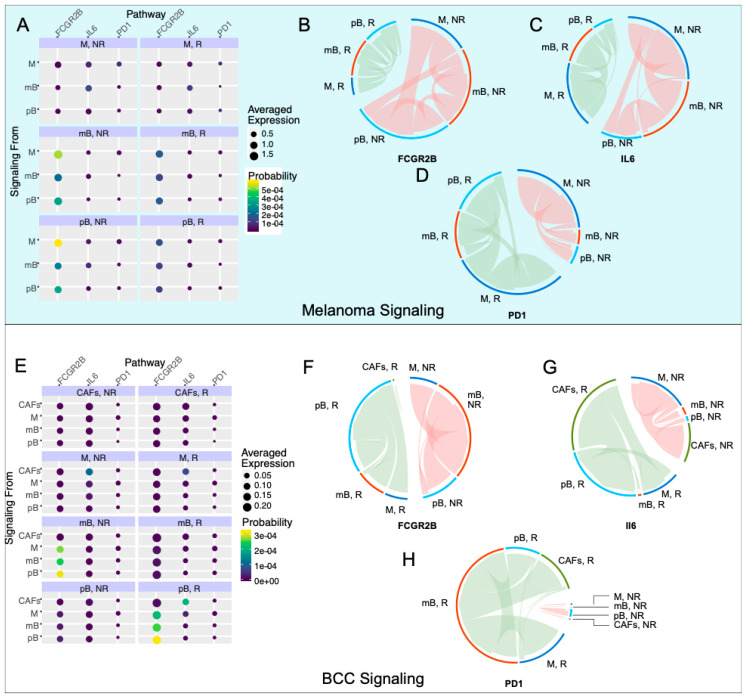
Inhibitory signaling is diminished in melanoma responders, whereas BCC responders experience increased inhibitory signaling. (**A**,**E**) Probability of signaling and averaged expression of the ligand/receptor/downstream target in each cell population for melanoma (**A**) and BCC (**E**). (**B**–**D**) Signaling of the Fc fragment of IgG receptor IIb (FCGR2B) (**B**), interleukin 6 (IL6) (**C**), PD1 (**D**) pathways for melanoma. (**F**–**H**) Signaling of the FCGR2B (**F**), IL6 (**G**), PD1 (**H**) pathways for BCC.

**Figure 5 cancers-12-02946-f005:**
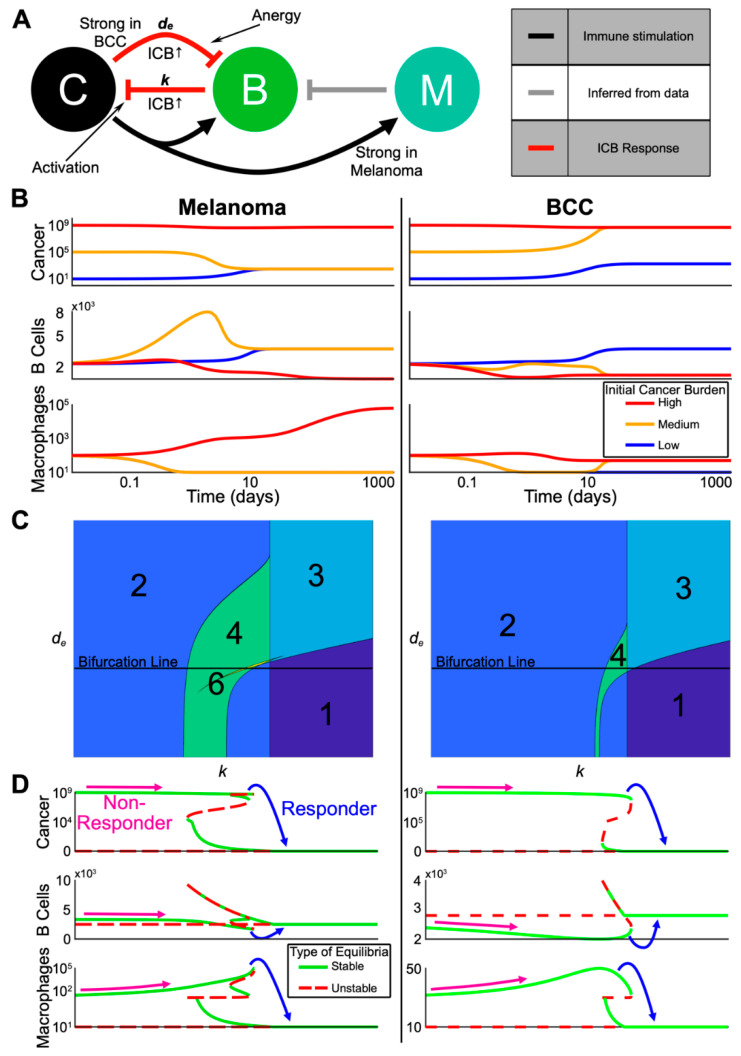
Multi-stability from complex interactions in melanoma and BCC can explain the heterogeneous response to ICB. (**A**) Schematic representation of the model (C = cancer, B = B cells, M = anti-inflammatory macrophages). Red arrows indicate processes assumed to be upregulated by checkpoint therapy. Grey arrow is inferred from our single-cell analysis. (**B**) Three trajectories with varying initial cancer population for each of melanoma and BCC. The color of the trajectory corresponds to the initial cancer population. All axes are log scale. (**C**) Contour plots showing the varying number of equilibria as the death rate of B cells (y-axis) and killing rate by B cells of cancer (x-axis) are varied. The “Bifurcation Line” indicates the values for which the bifurcations in (**D**) are plotted. (**D**) Bifurcation diagram in the killing rate, *k*, corresponding to the Bifurcation Line in (**C**). Possible starting and ending values for a non-responder and a responder are shown. Each Cancer and Macrophage axis is shown over two different scales for visualization purposes.

**Figure 6 cancers-12-02946-f006:**
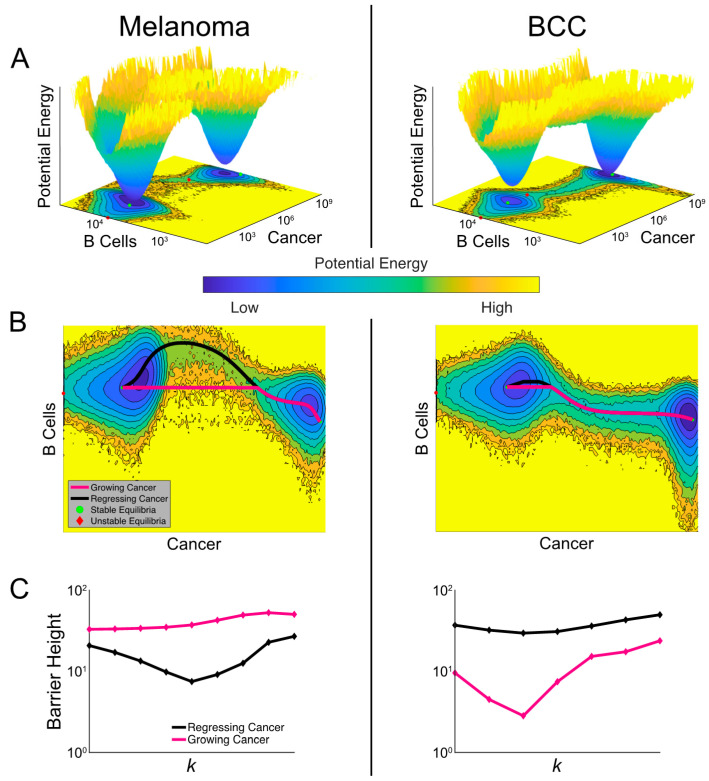
Comparisons of landscapes and transition paths for melanoma and BCC potentially explain the refractory response to checkpoint therapy in BCC. (**A**) Cancer-state energy landscape of both cancers with k = 1.8 × 10^−4^ and d_e_ = 1. Lower values indicate a higher probability of finding the system in that state. These values represent marginal potential energies, having marginalized over macrophages. A contour plot is shown below along with the ordinary differential equation (ODE)-determined equilibria, both stable (green dots) and unstable (red diamonds). (**B**) The transition paths between the stable equilibria plotted over the contour plots from (**A**). The black path is for regressing cancer and the magenta path is for growing cancer. (**C**) The barrier height between the two high-burden and low-burden cancer states as it varies with the killing rate k. The black curve shows the variation of the barrier height for regressing cancer and the magenta for growing cancer.

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
