# Peer review of "Divergent Resistance Mechanisms to Immunotherapy Explain Responses in Different Skin Cancers"

_cancers, 2020, doi:10.3390/cancers12102946_

Round 1
Reviewer 1 Report
In their manuscript ”Divergent resistance mechanisms to immunotherapy explains response in different skin cancers”, Dollinger et al. using transcriptomics from diagnostic basal cell carcinoma and malignant melanoma, try to infer that the number of B-cells and macrophages prior to immunotherapy correlates with responses.
While those findings corroborate previous reports on the role of tumour-associated macrophages and take up a new lead on the role of B-cells, the current manuscript falls short of providing a stratifying immune biomarker for immunotherapy.
Comments:
- There is a lack of rationale why the authors chose to compare a malignant matsatasting disease (melanoma) with a semi-malignant entitiy (BCC) that never matastasises, but can cause severe local infiltration. The main reason for systemic treatment of melanoma with immunotherapy is control of disseminated disease. It would thus have appeared more logically to compare localized to metastatic melanoma. The authors are therefore encouraged to include a data set on non-metastasisedd melanoma (even though those would not receive immunotherapy).
- The authors do not bother to mention that the composition of tumour-associated macrophages is very complex. It is well known that TAMs can exhibit both pro-tumour (M2 polarisation) and anti-tumour (M1 polarization) effects. This study therefore requires a subanalysis of the subtype of macrophages – which should be easily feasible given the availability of scRNASeq.
- No sub-analyses on well-known prognostic factors (e.g. BRAF statuts) are provided. This has to be added to be able to correlate the findings with known risk factors of malignant melanoma.
Reviewer 2 Report
Dollinger et. al have presented an interesting way to evaluating the predictors of response to immunotherapies in skin cancers and attempted to identify divergent mechanisms that confer resistance. While the methodological approach is interesting; the biological relevance is not adequately explained or even hypothesized. Additional, the roles of both B- cells as well as macrophages in determining the response to immunotherapies/checkpoint inhibition have previously been identified and discussed; hence the novelty of this manuscript is diminished. Lastly, there needs to be sound rationales for some of the specific assumptions made for the modelings/analyses. Some of my specific concerns are listed below:
- Revise lines 23-34. The statement here is opposite of the results presented with regards to B cells.
- The authors mention that the model presented here can predict the best ratio of macrophages to B cells; yet no such ratio is referenced here in the manuscript with the data.
- Introduction to checkpoint blockade immunotherapy needs to be improved and discussed further. It may be beneficial to also discuss how the checkpoint molecules affect the activities of B-cells and Macrophages and their interactions.
- What are the "cycling cells" mentioned in Figure 1? What is the biological significance or relevance of these cycling immune cells?
- Clarify sentences 88-89.
- For section 2.2, either present a table or list out the specific markers used for determination of activation versus anergic states.
- Add scale to figure 3E. It is difficult to interpret the results without it.
- Please explain why pro- versus anti-inflammatory macrophages' frequency was not stratified for responders versus non-responders (pre- and post-treatments)
- For figure 4, comment if inclusion of more markers could have improved the prediction of the cell-cell interactions/signaling.
- What are the "inhibitory macrophages" referenced in section 2.5? If cancer cells are included in the continnum dynamic model, please explain the rationale for leaving out other subsets of macrophages.
- With regards to the assumptions; what about the role of pro-inflammatory macrophages?
- "... we assumed that the cancer-mediated up-regulation of macrophage proliferation is weaker in BCC relative to melanoma." Please explain why you made this assumption? Cycling cells' frequency appear to be the same for both tumor types.
- Comment on the Tregs differences between responders and non responders between Melanoma and BCC.
Round 2
Reviewer 1 Report
The authors should be commended for addressing all of my comments with additional data, discussion and thoroughness.
Reviewer 2 Report
I thank that authors for taking time and doing an excellent job on addressing my comments with detailed explanations and additional data. I believe this has increased the quality of manuscript as well as improved ease of understanding by the potential readers. I have not further comments.